# Spatial and temporal regularization to estimate COVID-19 reproduction number *R(t)*: Promoting piecewise smoothness via convex optimization

**Patrice Abry**[1]*, **Nelly Pustelnik**[1], **Stéphane Roux**[1], **Pablo Jensen**[1,2], **Patrick Flandrin**[1], **Rémi Gribonval**[3], **Charles-Gérard Lucas**[1], **Éric Guichard**[2,4], **Pierre Borgnat**[1,2], **Nicolas Garnier**[1]

**1** Université de Lyon, ENS de Lyon, CNRS, Laboratoire de Physique, Lyon, France, **2** Université de Lyon, ENS de Lyon, CNRS, Inst. Systèmes Complexes, Lyon, France, **3** Univ Lyon, Inria, CNRS, ENS de Lyon, UCB Lyon 1, LIP UMR 5668, Lyon, France, **4** Université de Lyon, ENS de Lyon, CNRS, Laboratoire Triangle, Lyon, France

* patrice.abry@ens-lyon.fr

**Data Availability Statement:** Data are publicly documented from Health authority web sites. The three different URL from which the data used in the

## Abstract

Among the different indicators that quantify the spread of an epidemic such as the on-going COVID-19, stands first the reproduction number which measures how many people can be contaminated by an infected person. In order to permit the monitoring of the evolution of this number, a new estimation procedure is proposed here, assuming a well-accepted model for current incidence data, based on past observations. The novelty of the proposed approach is twofold: 1) the estimation of the reproduction number is achieved by convex optimization within a proximal-based inverse problem formulation, with constraints aimed at promoting piecewise smoothness; 2) the approach is developed in a multivariate setting, allowing for the simultaneous handling of multiple time series attached to different geographical regions, together with a spatial (graph-based) regularization of their evolutions in time. The effectiveness of the approach is first supported by simulations, and two main applications to real COVID-19 data are then discussed. The first one refers to the comparative evolution of the reproduction number for a number of countries, while the second one focuses on French departments and their joint analysis, leading to dynamic maps revealing the temporal co-evolution of their reproduction numbers.

## Introduction

### Context

The ongoing COVID-19 pandemic has produced an unprecedented health and economic crisis, urging for the development of adapted actions aimed at monitoring the spread of the new coronavirus. No country remained untouched, thus emphasizing the need for models and tools to perform quantitative predictions, enabling effective managements of patients or an

paper are downloaded are explicitly referred to in the manuscript. We also copy them here: https://coronavirus.jhu.edu/ https://github.com/CSSEGISandData/COVID-19/tree/master/csse_covid_19_data/csse_covid_19_time_series https://www.ecdc.europa.eu/ https://www.ecdc.europa.eu/sites/default/files/documents/COVID-19-geographic-disbtribution-worldwide.xlsx https://www.santepubliquefrance.fr/ https://www.data.gouv.fr/fr/datasets/.

**Funding:** The author(s) received no specific funding for this work.

**Competing interests:** The authors have declared that no competing interests exist.

optimized allocations of medical ressources. For instance, the outbreak of this unprecedented pandemic was characterized by a critical lack of tools able to perform predictions related to the pressure on hospital ressources (number of patients, masks, gloves, intensive care unit needs,. . .) [1, 2].

As a first step toward such an ambition goal, the present work focuses on the pandemic time evolution assessment. Indeed, all countries experienced a propagation mechanism that is basically universal in the onset phase: each infected person happened to infect in average more than one other person, leading to an initial exponential growth. The strength of the spread is quantified by the so-called *reproduction number* which measures how many people can be contaminated by an infected person. In the early phase where the growth is exponential, this is referred to as $R_0$ (for COVID-19, $R_0 \sim 3$ [3, 4]). As the pandemic develops and because more people get infected, the effective reproduction number evolves, hence becoming a function of time hereafter labeled $R(t)$. This can indeed end up with the extinction of the pandemic, $R(t) \to 0$, at the expense though of the contamination of a very large percentage of the total population, and of potentially dramatic consequences.

Rather than letting the pandemic develop until the reproduction number would eventually decrease below unity (in which case the spread would cease by itself), an active strategy amounts to take actions so as to limit contacts between individuals. This path has been followed by several countries which adopted effective *lockdown* policies, with the consequence that the reproduction number decreased significantly and rapidly, further remaining below unity as long as social distancing measures were enforced (see for example [4, 5]).

However, when lifting the lockdown is at stake, the situation may change with an expected increase in the number of inter-individual contacts, and monitoring in real time the evolution of the instantaneous reproduction number $R(t)$ becomes of the utmost importance: this is the core of the present work.

## Issues and related work

Monitoring and estimating $R(t)$ raises however a series of issues related to pandemic data modeling, to parameter estimation techniques and to data availability. Concerning the mathematical modeling of infectious diseases, the most celebrated approaches refer to *compartmental models* such as SIR ("Susceptible—Infectious—Recovered"), with variants such as SEIR ("Susceptible—Exposed—Infectious—Recovered"). Because such global models do not account well for spatial heterogeneity, clustering of human contact patterns, variability in typical number of contacts (cf. [6]), further refinements were proposed [7]. In such frameworks, the effective reproduction number at time $t$ can be inferred from a fit of the model to the data that leads to an estimated knowledge of the average of infecting contacts per unit time, of the mean infectious period, and of the fraction of the population that is still susceptible. These are powerful approaches that are descriptive and potentially predictive, yet at the expense of being fully parametric and thus requiring the use of dedicated and robust estimation procedures. Parameter estimation become all the more involved when the number of parameters grows and/or when the amount and quality of available data are low, as is the case for the COVID-19 pandemic *real-time* and *in emergency* monitoring.

Rather than resorting to fully parametric models and seeing $R(t)$ as the by-product of their identification, a more phenomenological, semi-parametric approach can be followed [8–10]. This approach has been reported as robust and potentially leading to relevant estimates of $R(t)$, even for epidemic spreading on realistic contact networks, where it is not possible to define a steady exponential growth phase and a basic reproduction number [6]. The underlying idea is to model incidence data $z(t)$ at time $t$ as resulting from a Poisson distribution with a time

evolving parameter adjusted to account for the data evolution, which depends on a function $\Phi(s)$ standing for the distribution of the *serial interval*. This function models the time between the onset of symptoms in a primary case and the onset of symptoms in secondary cases, or equivalently the probability that a person confirmed infected today was actually infected $s$ days earlier by another infected person. The serial interval function is thus an important ingredient of the model, accounting for the biological mechanisms in the epidemic evolution.

Assuming the distribution $\Phi$ to be known, the whole challenge in the actual use of the semi-parametric Poisson-based model thus consists in devising estimates $\hat{R}(t)$ of $R(t)$ with satisfactory statistical performance.

This has been classically addressed by approaches aimed at maximizing the likelihood attached to the model. This can be achieved, e.g., within several variants of Bayesian frameworks [5, 6, 8, 10], with even dedicated software packages (cf. e.g., https://shiny.dide.imperial. ac.uk/epiestim/). Instead, we promote here an alternative approach based on inverse problem formulations and proximal-operator based nonsmooth convex optimisation [11–15]. The questions of modeling and estimation, be they fully parametric or semi-parametric, are intimately intertwined with that of data availability. This will be further discussed but one can however remark at this point that many options are open, with a conditioning of the results to the choices that are made. There is first the nature of the incidence data used in the analysis (reported infected cases, hospitalizations, deaths) and the database they are extracted from. Next, there is the granularity of the data (whole country, regions, smaller units) and the specificities that can be attached to a specific choice as well as the comparisons that can be envisioned. In this respect, it is worth remarking that most analyses reported in the literature are based on (possibly multiple) univariate time series, whereas genuinely multivariate analyses (e.g., a joint analysis of the same type of data in different countries in order to compare health policies) might prove more informative.

## Goal, contributions and outlines

For that category of research work motivated by contributing *in emergency* to the societal stake of monitoring the pandemic evolution in *real-time*, or at least, on a *daily basis*, there are two classes of challenges: ensuring a robust and regular access to relevant data; rapidly developing analysis/estimation tools that are theoretically sound, practically usable on data actually available, and that may contribute to improving current monitoring strategies. In that spirit, the overarching goal of the present work is twofold: (1) proposing a new, more versatile framework for the estimation of $R(t)$ within the semi-parametric model of [8, 10], reformulating its estimation as an inverse problem whose functional is minimized by using non smooth proximal-based convex optimization; (2) inserting this approach in an extended multivariate framework, with applications to various complementary datasets corresponding to different geographical regions.

The paper is organized as follows. It first discusses data, as collected from different databases, with heterogeneity and uneven quality calling for some preprocessing that is detailed. In the present work, incidence data (thereafter labelled $z(t)$) refers to the number of daily new infections, either as reported in databases, or as *recomputed* from other available data such as hospitalization counts. Based on a semi-parametric model for $R(t)$, it is then discussed how its estimation can be phrased within a non smooth proximal-based convex optimization framework, intentionally designed to enforce piecewise linearity in the estimation of $R(t)$ via temporal regularization, as well as piecewise constancy in spatial variations of $R(t)$ by graph-based regularization. The effectiveness of these estimation tools is first illustrated on synthetic data, constructed from different models and simulating several scenarii, before being applied to

several real pandemic datasets. First, the number of daily new infections for many different countries across the world are analyzed independently. Second, focusing on France only, the number of daily new infections per continental France *départements* (*départements* constitute usual entities organizing the administrative life in France) are analyzed both independently and in a multivariate setting, illustrating the benefit of this latter formulation. Discussions, per-pectives and potential improvements are finally discussed.

## Materials and methods

### Data

**Datasets.**   In the present study, three sources of data were systematically used:

- **Source1(JHU)** *Johns Hopkins University* provides access to the cumulated daily reports of the number of infected, deceased and recovered persons, on a per country basis, for a large number of countries worldwide, essentially since inception of the COVID-19 crisis (January 1st, 2020). Data available at https://coronavirus.jhu.edu/ and https://raw.githubusercontent.com/CSSEGISandData/COVID-19/master/csse_covid_19_time_series/.

- **Source2(ECDPC)** The *European Centre for Disease Prevention and Control* provides similar information. Data available at https://www.ecdc.europa.eu/ and https://www.ecdc.europa.eu/sites/default/files/documents/COVID-19-geographic-disbtribution-worldwide.xlsx.

- **Source3(SPF)** *Santé-Publique-France* focuses on France only. It makes available on a daily basis a rich variety of pandemic-documented data across the France territory on a per *département*-basis, *départements* consisting of the usual granularity of geographical units (of roughly comparable sizes), used in France to address most administrative issues (an equivalent of counties for other countries). Source3(SPF) data are mostly based on hospital records, such as the daily reports of the number of currently hospitalized persons, together with the cumulated numbers of deceased and recovered persons with breakdowns by age and gender. Elementary algebra enables us to derive the daily number of new hospitalizations, used as a (delayed) proxy for daily new infections, assuming that a constant fraction of infected people is hospitalized. Data are however available only after March 20th, at https://www.santepubliquefrance.fr/ and https://www.data.gouv.fr/fr/datasets/.

**Time series.**   The data available on the different data repositories used here are strongly affected by outliers, which may stem from inaccuracy or misreporting in per country reporting procedures, or from changes in the way counts are collected, aggregated, and reported. In the present work, it has been chosen to preprocess data for outlier removal by applying to the raw time series a nonlinear filtering, consisting of a sliding-median over a 7-day window: outliers defined as ±2.5 standard deviation are replaced by window median to yield the pre-processed time series $z(t)$, from which the reproduction number $R(t)$ is estimated. An example of raw and pre-processed time series is illustrated in Fig 3.

When countries are studied independently, the estimation procedure is applied separately to each time series $z(t)$ of size $T$, the number of days available for analysis. When considering continental France *départements*, we are given $d$ time series $z_d(t)$ of size $T$ each, where $1 \leq d \leq D = 94$ indexes the *départements*. These time series are collected and stacked in a matrix of size $D \times T$, and they analyzed both independently and jointly.

### Model and estimation procedures

**Model.**   Although they can be used for envisioning the impact of possible scenarii in the future development of an on-going epidemic [3], SIR models, because they require the full

estimation of numerous parameters, are often used a posteriori (e.g., long after the epidemic) with consolidated and accurate datasets. During the spread phase and in order to account for the on-line/on-the-fly need to monitor the pandemic and to offer some robustness to partial/incomplete/noisy data, less detailed semi-parametric models focusing on the only estimation of the time-dependent reproduction number can be preferred [8, 9, 16].

Let $R(t)$ denote the instantaneous reproduction number to be estimated and $z(t)$ be the number of daily new infections. It has been proposed in [8, 10] that $\{z(t), t = 1, \ldots, T\}$ can be modeled as a nonstationary time series consisting of a collection of random variables, each drawn from a Poisson distribution $\mathcal{P}_{p_t}$ whose parameter $p_t$ depends on the past observations of $z(t)$, on the current value of $R(t)$, and on the *serial interval* function $\Phi(\cdot)$:

$$\mathbb{P}(z(1), \ldots, z(t)) = \prod_{s=1}^{t} \mathcal{P}_{p_s}(z(s)), \text{ with } p_s = R(s) \sum_{u \geq 1} \Phi(u) z(s - u). \tag{1}$$

The *serial interval* function $\Phi(\cdot)$ constitutes a key ingredient of the model, whose importance and role in pandemic evolution has been mentioned in Introduction. It is assumed to be independent of calendar time (i.e., constant across the epidemic outbreak), and, importantly, independent of $R(t)$, whose role is to account for the time dependencies in pandemic propagation mechanisms.

For the COVID-19 pandemic, several studies have empirically estimated the serial interval function $\Phi(\cdot)$ [17, 18]. For convenience, $\Phi(\cdot)$ has been modeled as a Gamma distribution, with shape and rate parameters 1.87 and 0.28, respectively (corresponding to mean and standard deviations of 6.6 and 3.5 days, see [5] and references therein). These choices and assumptions have been followed and used here, and the corresponding function is illustrated in Fig 1.

In essence, the model in Eq (1) is *univariate* (only one time series is modeled at a time), and based on a Poisson marginal distribution. It is also *nonstationary*, as the Poisson rate evolves along time. The key ingredient of this model consists of the Poisson rate evolving as a weighted moving average of past observations, which is qualitatively based on the following rationale: when

$$\hat{R}_{\text{naive}}(t) = z(t) / \sum_{s \geq 1} \Phi(s) z(t - s) \tag{2}$$

is above 1, the epidemic is growing and, conversely, when this ratio is below 1, it decreases and eventually vanishes.

**Non-smooth convex optimisation.** The whole challenge in the actual use of the semi-parametric Poisson-based model described above thus consists in devising estimates $\hat{R}(t)$ of $R(t)$ that have better statistical performance (more robust, reliable, and hence usable) than the direct brute-force and naive form defined in Eq 2. To estimate $R(t)$, and instead of using Bayesian frameworks that are considered state-of-the-art tools for epidemic evolution analysis, we propose and promote here an alternative approach based on an inverse problem formulation. Its main principle is to assume some form of *temporal regularity* in the evolution of $R(t)$ (we use a piecewise linear model in the following). In the case of a joint estimation of $R(t)$ across several continental France *départements*, we further assume some form of *spatial regularity*, i.e., that the values of $R(t)$ for neighboring *départements* are similar.

*Univariate setting.* For a single country, or a single *département*, the observed (possibly pre-processed) data $\{z(t), 1 \leq t \leq T\}$ is represented by a $T$-dimensional vector $\mathbf{z} \in \mathbb{R}^T$. Recalling that the Poisson law is $\mathbb{P}(Z = n | p) = \frac{p^n}{n!} e^{-p}$ for each integer $n \geq 0$, the negative log-likelihood

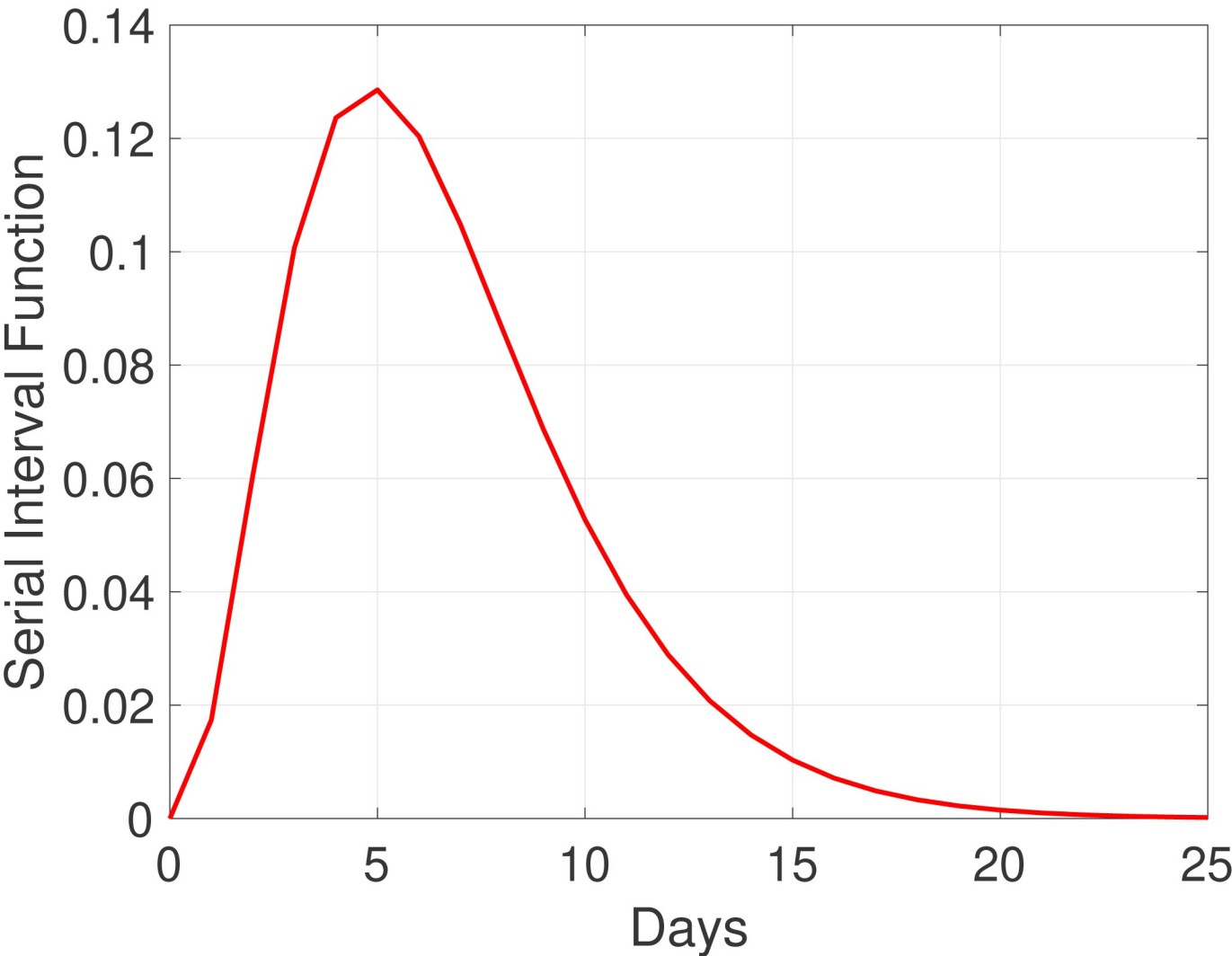

**Fig 1. Serial interval function Φ modeled as a Gamma distribution with mean and standard deviation of 6.6 and 3.5 days, following [17].**

of observing $\mathbf{z}$ given a vector $\mathbf{p} \in \mathbb{R}^T$ of Poisson parameters $p_t$ is

$$-\log \mathbb{P}(\mathbf{z} \mid \mathbf{p}) = \sum_{t=1}^{T} - \log \mathbb{P}(Z(t) = z(t) \mid p_t) = \sum_{t=1}^{T}(p_t - z(t) \log p_t + \log [z(t)!]), \quad (3)$$

where $\mathbf{r} \in \mathbb{R}^T$ is the (unknown) vector of values of $R(t)$. Up to an additive term independent of $\mathbf{p}$, this is equal to the KL-divergence (cf. Section 5.4. in [15]):

$$D_{\mathrm{KL}}(\mathbf{z} \mid \mathbf{p}) = \sum_{t=1}^{T}\left( z(t) \log \frac{z(t)}{p_t} + p_t - z(t) \right). \quad (4)$$

Given the vector of observed values $\mathbf{z}$, the serial interval function $\Phi(\cdot)$, and the number of days $T$, the vector $\mathbf{p}$ given by (1) reads $\mathbf{p} = \mathbf{r} \odot \Phi\mathbf{z}$, with $\odot$ the entrywise product and $\Phi \in \mathbb{R}^{T \times T}$ the matrix with entries $\Phi_{ij} = \Phi(i - j)$.

Maximum likelihood estimation of $\mathbf{r}$ (i.e., minimization of the negative log-likelihood) leads to an optimization problem $\min_\mathbf{r} D_{\mathrm{KL}}(\mathbf{z}|\mathbf{r} \odot \Phi\mathbf{z})$ which does not ensure any regularity of $R(t)$. To ensure *temporal* regularity, we propose a penalized approach using $\hat{\mathbf{r}} = \mathrm{argmin}_\mathbf{r} D_{\mathrm{KL}}(\mathbf{z} \mid \mathbf{r} \odot \Phi\mathbf{z}) + \Omega(\mathbf{r})$ where $\Omega$ denotes a penalty function.

Here we wish to promote a piecewise affine and continuous behavior, which may be accomplished [19, 20] using $\Omega(\mathbf{r}) = \lambda_{\mathrm{time}}\|\mathbf{D}_2\,\mathbf{r}\|_1$, where $\mathbf{D}_2$ is the matrix associated with a Laplacian filter (second order discrete temporal derivatives), $\|\cdot\|_1$ denotes the $\ell_1$-norm (i.e., the sum of the absolute values of all entries), and $\lambda_{\mathrm{time}}$ is a penalty factor to be tuned. This leads to the following optimization problem:

$$\hat{\mathbf{r}} = \arg\min_\mathbf{r} D_{\mathrm{KL}}(\mathbf{z} \mid \mathbf{r} \odot \Phi\mathbf{z}) + \lambda_{\mathrm{time}}\|\mathbf{D}_2\mathbf{r}\|_1. \tag{5}$$

*Spatially regularized setting.* In the case of multiple *départements*, we consider multiple vectors $(\mathbf{z}_d \in \mathbb{R}^T, 1 \le d \le D)$ associated to the $D$ time series, and multiple vectors of unknown $(\mathbf{r}_d \in \mathbb{R}^T, 1 \le d \le D)$, which can be gathered into matrices: a data matrix $\mathbf{Z} \in \mathbb{R}^{T \times D}$ whose columns are $\mathbf{z}_d$ and a matrix of unknown $\mathbf{R} \in \mathbb{R}^{T \times D}$ whose columns are the quantities to be estimated $\mathbf{r}_d$.

A first possibility is to proceed to independent estimations of the $(\mathbf{r}_d \in \mathbb{R}^T, 1 \le d \le D)$ by addressing the separate optimization problems

$$\hat{\mathbf{r}}_d = \arg\min_\mathbf{r} D_{\mathrm{KL}}(\mathbf{z}_d \mid \mathbf{r} \odot \Phi\mathbf{z}_d) + \lambda_{\mathrm{time}}\|\mathbf{D}_2\mathbf{r}\|_1,$$

which can be equivalently rewritten into a matrix form:

$$\hat{\mathbf{R}}_{\mathrm{indep}} = \arg\min_\mathbf{r} D_{\mathrm{KL}}(\mathbf{Z} \mid \mathbf{R} \odot \Phi\mathbf{Z}) + \lambda_{\mathrm{time}}\|\mathbf{D}_2\mathbf{R}\|_1, \tag{6}$$

where $D_{\mathrm{KL}}(\mathbf{Z} \mid \mathbf{R} \odot \Phi\mathbf{Z}) := \sum_{d=1}^D D_{\mathrm{KL}}(\mathbf{z}_d \mid \mathbf{r}_d \odot \Phi\mathbf{z}_d)$, and $\|\mathbf{D}_2\mathbf{R}\|_1 = \sum_{d=1}^D \|\mathbf{D}_2\mathbf{r}_d\|_1$ is the entry-wise $\ell^1$ norm of $\mathbf{D}_2\,\mathbf{R}$, i.e., the sum of the absolute values of all its entries.

An alternative is to estimate jointly the $(\mathbf{r}_d \in \mathbb{R}^T, 1 \le d \le D)$ using a penalty function promoting *spatial* regularity. To account for spatial regularity, we use a spatial analogue of $\mathbf{D}_2$ promoting spatially piecewise constant solutions. The $D$ continental France *départements* can be considered as the vertices of a graph, where edges are present between adjacent *départements*.

From the adjacency matrix $\mathbf{A} \in \mathbb{R}^{D \times D}$ of this graph ($\mathbf{A}_{ij} = 1$ if there is an edge $e = (i, j)$ in the graph, $\mathbf{A}_{ij} = 0$ otherwise), the global variation of the function on the graphs can be computed as $\sum_{ij} \mathbf{A}_{ij}(\mathbf{R}_{ti} - \mathbf{R}_{tj})^2$ and it is known that this can be accessed through the so-called (combinatorial) Laplacian of the graph: $\mathbf{L} = \Delta - \mathbf{A}$ where $\Delta$ is the diagonal matrix of the degrees ($\Delta_{ii} = \sum_j \mathbf{A}_{ij}$) [21]. However, in order to promote smoothness over the graph while keeping some sparse discontinuities on some edges, it is preferable to regularize using a Total Variation on the graph, which amounts to take the $\ell_1$-norm of these gradients $(\mathbf{R}_{ti} - \mathbf{R}_{tj})$ on all existing edges. For that, let us introduce the incidence matrix $\mathbf{B} \in \mathbb{R}^{E \times D}$ such that $\mathbf{L} = \mathbf{B}^\top \mathbf{B}$ where $E$ is the number of edges and, on each line representing an existing edge $e = (i, j)$, we set $\mathbf{B}_{e,i} = 1$ and $\mathbf{B}_{e,j} = -1$. Then, the $\ell_1$-norm $\|\mathbf{R}\mathbf{B}^\top\|_1 = \|\mathbf{B}\mathbf{R}^\top\|_1$ is equal to $\sum_{t=1}^T \sum_{(i,j):\mathbf{A}_{ij}=1} |\mathbf{R}_{ti} - \mathbf{R}_{tj}|$. Alternatively, it can be computed as $\|\mathbf{R}\mathbf{B}^\top\|_1 = \sum_{t=1}^T \|\mathbf{B}\mathbf{r}(t)\|_1$ where $\mathbf{r}(t) \in \mathbb{R}^D$ is the $t$-th row of $\mathbf{R}$, which gathers the values across all *départements* at a given time $t$. From that, we can define the regularized optimization problem:

$$\hat{\mathbf{R}}_{\mathrm{joint}} = \arg\min_\mathbf{r} D_{\mathrm{KL}}(\mathbf{Z} \mid \mathbf{R} \odot \Phi\mathbf{Z}) + \lambda_{\mathrm{time}}\|\mathbf{D}_2\mathbf{R}\|_1 + \lambda_{\mathrm{space}}\|\mathbf{R}\mathbf{B}^\top\|_1. \tag{7}$$

Optimization problems (6) and (7) involve convex, lower semi-continuous, proper and non-negative functions, hence their set of minimizers is non-empty and convex [11]. We will discuss right after how to compute these using proximal algorithms.

By the known sparsity-promoting properties of $\ell^1$ regularizers and their variants, the corresponding solutions are such that $\mathbf{D}_2\,\mathbf{R}$ and/or $\mathbf{RB}^\top$ are sparse matrices, in the sense that these matrices of (second order temporal or first order spatial) derivatives have many zero entries. The higher the penalty factors $\lambda_{\text{time}}$ and $\lambda_{\text{space}}$, the more zeroes in these matrices. In particular, when $\lambda_{\text{space}} = 0$, no spatial regularization is performed and (7) is equivalent to (6). When $\lambda_{\text{space}}$ is large enough, $\mathbf{RB}^\top$ is *exactly* zero, which implies that $\mathbf{r}(t)$ is constant at each time since the graph of *départements* is connected.

**Optimization using a proximal algorithm.** The considered optimization problems are of the form

$$\text{minimize}_\mathbf{R}\,\Psi(\mathbf{R}) := F(\mathbf{R}) + \sum_{m=1}^{M} G_m(K_m(\mathbf{R})), \tag{8}$$

where $F$ and $G_m$ are proper lower semi-continuous convex, and $K_m$ are bounded linear operators. A classical case for $m = 1$ is typically addressed with the Chambolle-Pock algorithm [22], which has been recently adapted for multiple regularization terms as in Eq. 8 of [23]. To handle the lack of smoothness of Lipschitz differentiability for the considered functions $F$ and $G_m$, these approaches rely on their proximity operators. We recall that the proximity operator of a convex, lower semi-continuous function $\varphi$ is defined as [24]

$$\mathrm{prox}_\varphi(\mathbf{y}) = \arg\min_\mathbf{x} \frac{1}{2}\|\mathbf{y} - \mathbf{x}\|_2^2 + \varphi(\mathbf{x}).$$

In our case, we consider a separable data fidelity term:

$$F(\mathbf{R}) = D_{\text{KL}}(\mathbf{Z} \mid \mathbf{R} \odot \mathbf{\Phi Z}) = \sum_{td}\left[ \mathbf{R}_{td}\cdot(\mathbf{\Phi Z})_{td} - \mathbf{Z}_{td} + \mathbf{Z}_{td}\log\left(\frac{\mathbf{Z}_{td}}{\mathbf{R}_{td}\cdot(\mathbf{\Phi Z})_{td}}\right)\right]. \tag{9}$$

As this is a separable function of the entries of its input, its associated proximity operator can be computed component by component [25]:

$$(\mathrm{prox}_{\tau F}(\mathbf{X}))_{td} = \frac{\mathbf{X}_{td} - \tau\cdot(\mathbf{\Phi Z})_{td} + \sqrt{|\mathbf{X}_{td} - \tau\cdot(\mathbf{\Phi Z})_{td}|^2 + 4\tau\mathbf{Z}_{td}}}{2},$$

where $\tau > 0$.

We further consider $G_m(\cdot) = \|.\|_1$, $m = 1, 2$, and $K_1(\mathbf{R}) := \lambda_{\text{time}}\,\mathbf{D}_2\,\mathbf{R}$, $K_2(\mathbf{R}) := \lambda_{\text{space}}\,\mathbf{RB}^\top$. The proximity operators associated to $G_m$ read:

$$(\mathrm{prox}_{\tau\|.\|_1}(\mathbf{X}))_{td} = \left(1 - \frac{\tau}{|\mathbf{X}_{td}|}\right)_+ \mathbf{X}_{td},$$

where $(.)_+ = \max(0,.)$. In Algorithm 1, we express explicitly Algorithm 161 of [23] for our setting, considering the Moreau identity that provides the relation between the proximity operator of a function and the proximity operator of its conjugate (cf. Eq. (8) of [23]). The choice of the parameters $\tau$ and $\sigma_m$ impacts the convergence guarantees. In this work, we adapt a standard choice provided by [22] to this extended framework. The adjoint of $K_m$, denoted $K_m^*$, is given by $K_1^*(\mathbf{Y}) := \lambda_{\text{time}}\mathbf{D}_2^\top\mathbf{Y}$, $K_2^*(\mathbf{Y}) := \lambda_{\text{space}}\mathbf{YB}$. The sequence $(\mathbf{R}^{(k+1)})_{k\in\mathbb{N}}$ converges to a minimizer of (7) (cf. Thm 8.2 of [23]).

**Algorithm 1**: Chambolle-Pock with multiple penalization terms

```
Input: data Z, tolerance ε > 0
```
**Initialization:,** $k = 0,\ \tau = \sigma_m = 0.99/\sqrt{\sum_{m=1,2}\|K_m\|^2}$

$\mathbf{R}^{(0)} = \mathbf{Z},\ \mathbf{Y}_m^{(0)} = K_m(\mathbf{R}^{(0)})$

**while** $|\Psi(\mathbf{R}^{(k+1)}) - \Psi(\mathbf{R}^{(k)})|/\Psi(\mathbf{R}^{(k)}) > \epsilon$ **do**

  **for** $m = 1, 2$ **do**

   $\mathbf{Y}_m^{(k+1)} = \mathbf{Y}_m^{(k)} + \sigma_m K_m(\mathbf{R}^{(k)}) - \sigma_m \mathrm{prox}_{\frac{1}{\sigma_m}G_m}\left(\frac{1}{\sigma_m}\mathbf{Y}_m^{(k)} + K_m(\mathbf{R}^{(k)})\right);$

   $\mathbf{R}^{(k+1)} = \mathrm{prox}_{\tau F}(\mathbf{R}^{(k)} - \tau \sum_m K_m^*(2\mathbf{Y}_m^{(k+1)} - \mathbf{Y}_m^{(k)}));$

   $k \leftarrow k + 1;$

**Result**: $\mathbf{R}^{(\mathrm{end})}$

## Results

### Synthetic data

To assess the relevance and performance of the proposed estimation procedure detailed above, it is first applied to two different synthetic time series $z(t)$. The first one is synthesized using directly the model in Eq (1), with the same serial interval function $\Phi(t)$ as that used for the estimation, and using an a priori prescribed function $R(t)$. The second one is produced from solving a compartmental (SIR type) model. For such models, $R(t)$ can be theoretically related to the time scale parameters entering their definition, as the ratio between the infection time scale and the quitting infection (be it by death or recovery) time scale [26, 27]. The theoretical serial function $\Phi$ associated to that model and to its parameters is computed analytically (cf., e.g., [28]) and used in the estimation procedure.

For both cases, the same a priori prescribed function $R(t)$, to be estimated, is chosen as constant ($R = 2.2$) over the first 45 days to model the epidemic outbreak, followed by a linear decrease (till below 1) over the next 45 days to model lockdown benefits, with finally an abrupt linear increase for the last 10 days, modeling a possible outbreak at when lockdown is lifted. Additive Gaussian noise is superimposed to the data produced by the models to account for outliers and misreporting.

For both cases, the proposed estimation procedure (obtained with $\lambda_{\mathrm{time}}$ set to the same values as those used to analyze real data in the next section) outperforms the naive estimates (2), which turn out to be very irregular (cf. Fig 2). The proposed estimates notably capture well the three different phases of $R(t)$ (stable, decreasing and increasing), with notably a rapid and accurate reaction to the increasing change in the 10 last days.

### COVID-19 data

The present section aims to apply the model and estimation tools proposed above to actual COVID-19 data. First, specific methodological issues are addressed, related to tuning the hyperparameter(s) $\lambda_{\mathrm{time}}$ or ($\lambda_{\mathrm{time}}$, $\lambda_{\mathrm{space}}$) in univariate and multivariate settings, and to comparing the consistency between different estimates of $R(t)$ obtained from the same incidence data, yet downloaded from different repositories. Then, the estimation tools are applied to the estimation of $R(t)$, both independently for numerous countries and jointly for the 94 continental France *départements*.

Estimation of $R(t)$ is performed daily, with $T$ thus increasing every day, and updated results are uploaded on a regular basis on a dedicated webpage (cf. http://perso.ens-lyon.fr/patrice. abry).

**Regularization hyperparameter tuning.**   A critical issue associated with the practical use of the estimates based on the optimization problems (5) and (7) lies in the tuning of the

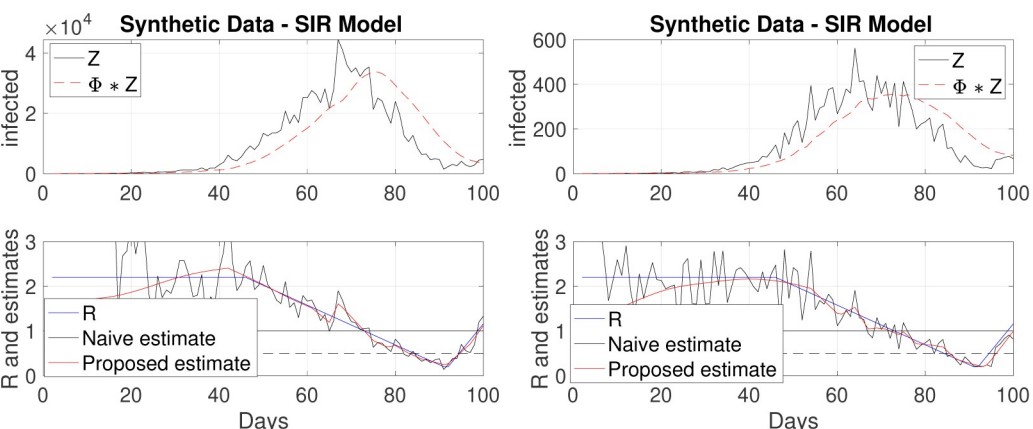

**Fig 2. Estimated reproduction numbers $R(t)$ on synthetic data, produced by the Poisson model (1) (left) and by a SIR model (right).** The true $R(t)$ (blue line) is piecewise linear: constant till day 45, decreasing till day 90 and increasing for the last 10 days. The proposed estimate (red) performs better than the naive estimate (black) (cf. Eq (2)) and detects well the changes, notably it quickly reacts to the increase of the last 10 days.

hyperparameters balancing data fidelity terms and penalization terms. While automated and data-driven procedures can be devised, following works such as [29] and references therein, let us analyze the forms of the functional to be minimized, so as to compute relevant orders of magnitude for these hyperparameters.

Let us start with the univariate estimation (5). Using $\lambda_{\text{time}} = 0$ implies no regularization and the achieved estimate turns out to be as *noisy* as the one obtained with a naive estimator (cf. Eq (2)). Conversely, for large enough $\lambda_{\text{time}}$, the proposed estimate becomes *exactly* a constant, missing any time evolution. Tuning $\lambda_{\text{time}}$ is thus critical but can become tedious, especially because differences across countries (or across *départements* in France) are likely to require different choices for $\lambda_{\text{time}}$. However, a careful analysis of the functional to minimize shows that the data fidelity term (9), based on a Kullback-Leibler divergence, scales proportionally to the input incidence data $z$ while the penalization term, based on the regularization of $R(t)$, is independent of the actual values of $z$. Therefore, the same estimate for $R(t)$ is obtained if we replace $z$ with $\alpha \times z$ and $\lambda$ with $\alpha \times \lambda$. Because orders of magnitude of $z$ are different amongst countries (either because of differences in population size, or of pandemic impact), this critical observation leads us to apply the estimate not to the raw data $z$ but to a normalized version $z/\text{std}(z)$, alleviating the burden of selecting one $\lambda_{\text{time}}$ per country, instead enabling to select one same $\lambda_{\text{time}}$ for all countries and further permitting to compare the estimated $R(t)$'s across countries for equivalent levels of regularization.

Considering now the graph-based spatially-regularized estimates (7) while keeping fixed $\lambda_{\text{time}}$, the different $R(t)$ are analyzed independently for each *département* when $\lambda_{\text{space}} = 0$. Conversely, choosing a large enough $\lambda_{\text{space}}$ yields *exactly identical* estimates across *départments* that are, satisfactorily, very close to what is obtained from data aggregated over France prior to estimation. Further, the connectivity graph amongst the 94 continental France *départements* leads to an adjacency matrix with 475 non-zero off-diagonal entries (set to the value 1), associated to as many edges as existing in the graph. Therefore, a careful examination of (7) shows that the spatial and temporal regularizations have equivalent weights when $\lambda_{\text{time}}$ and $\lambda_{\text{time}}$ are chosen such that

$$94 \times \lambda_{\text{time}} = 2 \times 475 \times \lambda_{\text{space}}. \tag{10}$$

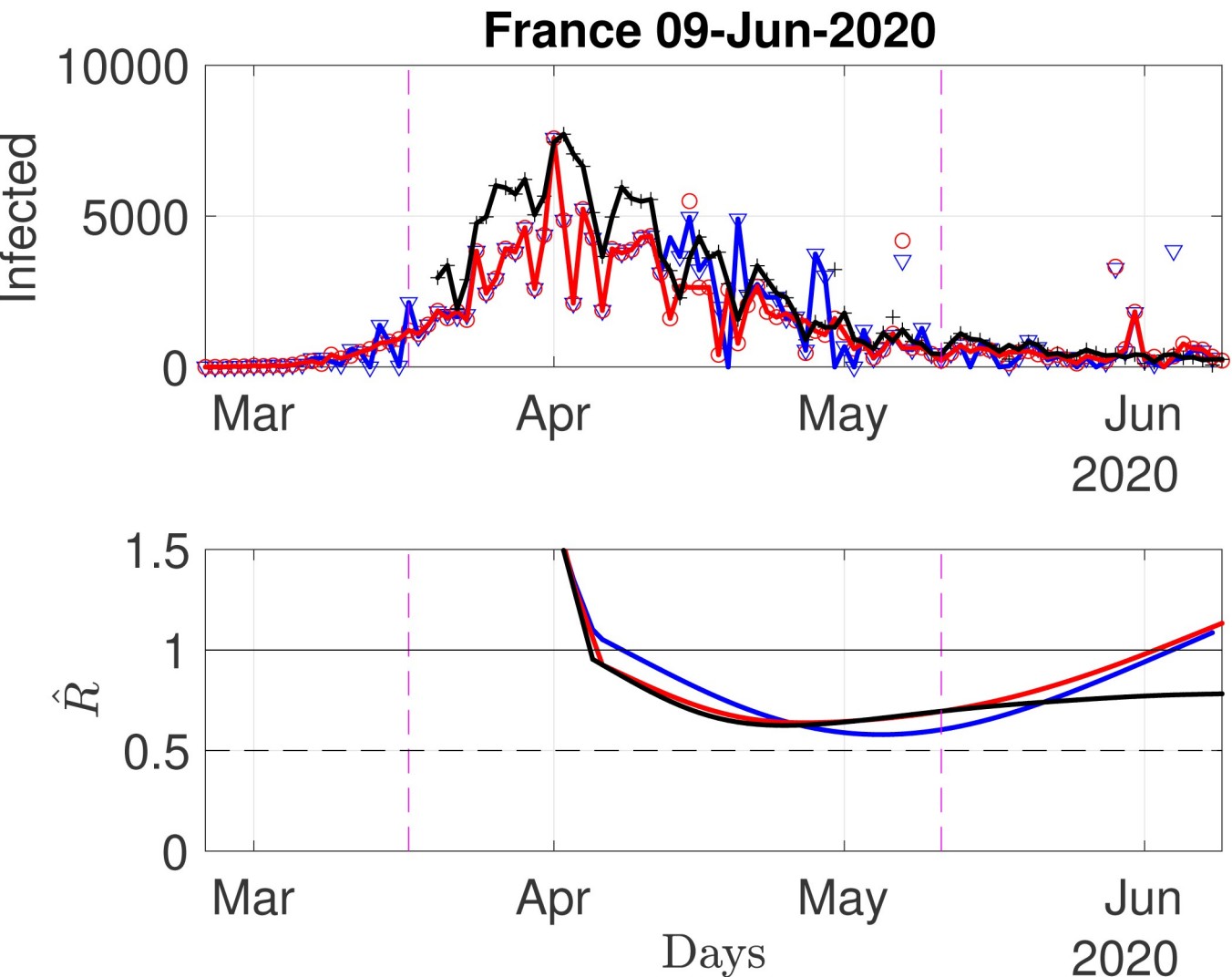

**Fig 3. Daily new confirmed cases for France, from three different sources.** Top row: raw data (symbols) and sliding median preprocessed data (connected lines) from Source1(JHU) (blue) and Source2(ECDPC)(red) and Source3(SPF) (black). Bottom row: corresponding estimates of $R(t)$.

The use of $z/\text{std}(z)$ and of (10) above gives a relevant first-order guess to the tuning of $\lambda_{\text{time}}$ and of $(\lambda_{\text{time}}, \lambda_{\text{space}})$.

**Estimate consistency using different repository sources.** When undertaking such work dedicated to on-going events, to daily evolutions, and to a real stake in forecasting future trends, a solid access to reliable data is critical. As previously mentioned, three sources of data are used, each including data for France, which are thus now used to assess the impact of data sources on estimated $R(t)$. Source1(JHU) and Source2(ECDPC) provide cumulated numbers of confirmed cases counted at national levels and (in principle) including all reported cases from any source (hospital, death at home or in care homes. . .). Source3(SPF) does not report that same number, but a collection of other figures related to hospital counts only, from which a daily number of new hospitalizations can be reconstructed and used as a proxy for daily new infections. The corresponding raw and (sliding-median) preprocessed data, illustrated in Fig 3, show overall comparable shapes and evolutions, yet with clearly visible discrepancies of two kinds.

First, Source1(JHU) and Source2(ECDPC), consisting of crude reports of number of confirmed cases are prone to outliers. Those can result from miscounts, from pointwise incorporations of new figures, such as the progressive inclusion of cases from *EHPAD* (care homes) in France, or from corrections of previous erroneous reports. Conversely, data from Source3 (SPF), based on hospital reports, suffer from far less outliers, yet at the cost of providing only partial figures.

Second, in France, as in numerous other countries worldwide, the procedure on which confirmed case counts are based, changed several times during the pandemic period, yielding possibly some artificial increase in the local average number of daily new confirmed cases. This has notably been the case for France, prior to the end of the lockdown period (mid-May), when the number of tests performed has regularly increased for about two weeks, or more recently early June when the count procedures has been changed again, likely because of the massive use of serology tests. Because the estimate of $R(t)$ essentially relies on comparing a daily number against a past moving average, these changes lead to significant biases that cannot be easily accounted for, but vanishes after some duration controlled by the typical width of the serial distribution $\Phi$ (of the order of ten days).

Fig 3 further compares, for a relevant value of $\lambda_{time}$ ($\lambda_{time} = 50$, see above), estimates obtained from the three different sources of data. For Source1(JHU) and Source2(ECDCP), overall shapes in the time evolution of estimates are consistent, showing a mild yet clear increase of $R(t)$ for the period ranging from early May to May 20th, likely corresponding to a bias induced by the regular increase of tests actually performed in France. This recent increase is however not seen in Source3(SPF)-based estimates that remain very stable. These comparisons however also clearly show that estimates are impacted by outliers and thus do depend on preprocessing. These considerations led to the final choice, used hereafter, of a threshold of ±2.5 std, in the sliding median denoising.

**Confirmed infection cases across the world.**  To report estimated $R(t)$'s for different countries, data from Source2(ECDPC) are used as they are of better quality than data from Source1(JHU), and because hospital-based data (as in Source3(SPF)) are not easily available for numerous different countries. Visual inspection led us to choose, uniformly for all countries, two values of the temporal regularization parameter: $\lambda_{time} = 50$ to produce a strongly-regularized, hence slowly varying estimate, and $\lambda_{time} = 3.5$ for a milder regularization, and hence a more reactive estimate. These estimates being by construction designed to favor piecewise linear behaviors, local trends can be estimated by computing (robust) estimates of the derivatives $\hat{\beta}(t)$ of $\hat{R}(t)$. The slow and less slow estimates of $\hat{R}(t)$ thus provide a slow and less slow estimate of the local trends. Intuitively, these local trends can be seen as predictors for the forthcoming value of $R$: $\hat{R}(t + n) = \hat{R}(t) + n\hat{\beta}(t)$.

Let us start by inspecting again data for France, further comparing estimates stemming from data in Source2(ECDPC) or in Source3(SPF) (cf. Fig 4). As discussed earlier, data from Source2(ECDPC) show far more outliers that data from Source3(SPF), thus impacting estimation of $R$ and $\beta$. As expected, the strongly regularized estimates ($\lambda_{time} = 50$) are less sensitive than the less regularized ones ($\lambda_{time} = 3.5$), yet discrepancies in estimates are significant, as data from Source2(ECDPC) yields, for June 9th, estimates of $R$ slightly above 1, while that from Source3(SPF) remain steadily around 0.7, with no or mild local trends. Again, this might be because late May, France has started massive serology testing, mostly performed outside hospitals. This yielded an abrupt increase in the number of new confirmed cases, biasing upward the estimates of $R(t)$. However, the short-term local trend for June 9th goes also downward, suggesting that the model is incorporating these irregularities and that estimates will return to unbiased after an estimation time controlled by the typical width of the serial

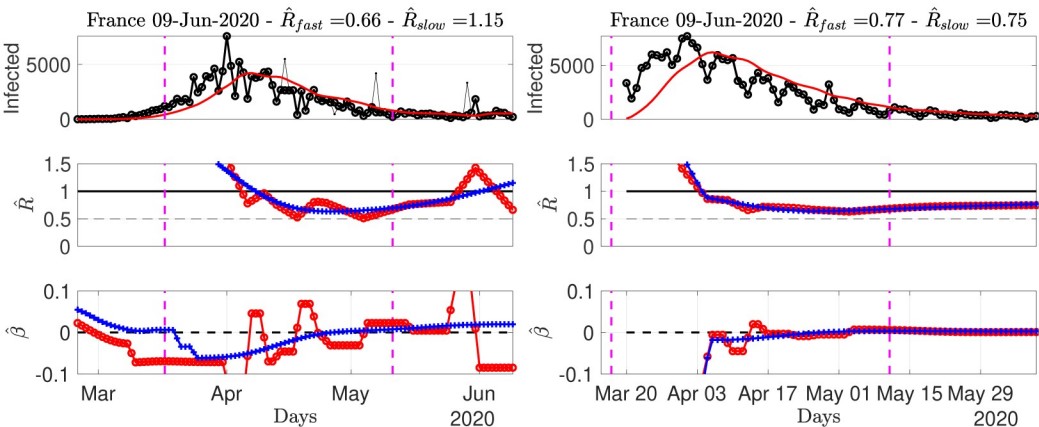

**Fig 4. Number of daily new confirmed cases for France, reproduction numbers and local trends, using data from Source2(ECDPC) (left) and Source3(SPF) (right) (reconstructed proxy from hospital counts).** Top: time series. Middle: fast (red) and slowly (blue) evolving estimates of $R(t)$. Bottom: fast (red) and slowly (blue) evolving estimates of local trends $\beta(t)$. The title of the plots reports the slow and fast estimates of $R$ for the current day.

distribution $\Phi$ (of the order of ten days). This recent increase is not seen in Source3(SPF)-based estimates that remain very stable, potentially suggesting that hospital-based data are much less affected by changes in testing policies.

This local analysis at the current date can be complemented by a more global view on what happened since the lifting of the lockdown. Considering the whole period starting from May 11th we end up with triplets [5th percentile; **median**; 95th percentile] that read as given in Table 1:

Source2(ECDPC) provides data for several tens of countries. Figs 5 to 8 report $\hat{R}(t)$ and $\hat{\beta}(t)$ for several selected countries. More figures are available at perso.ens-lyon.fr/patrice.abry. As of June 9th (time of writing), Fig 5 shows that, for most European countries, the pandemic seems to remain under control despite lifting of the lockdown, with (slowly varying) estimates of $R$ remaining stable below 1, ranging from 0.7 to 0.8 depending on countries, and (slowly varying) trends around 0. Sweden and Portugal (not shown here) display less favorable patterns, as well as, to a lesser extent, The Netherlands, raising the question of whether this might be a potential consequence of less stringent lockdown rules compared to neighboring European countries.

Fig 6 shows that while $\hat{R}$ for Canada is clearly below 1 since early May, with a negative local trend, the USA are still bouncing back and forth around 1. South America is in the above 1 phase but starts to show negative local trends. Fig 7 indicates that Iran, India or Indonesia are in the critical phase with $\hat{R}(t) > 1$. Fig 8 shows that data for African countries are uneasy to analyze, and that several countries such as Egypt or South Africa are in pandemic growing phases.

**Table 1. Triplets [5th percentile; median; 95th percentile] for the average of the reproduction number in France from May 11th to June 9th.**

|  | Source2(ECDC) | Source3(SPF) |
|---|---|---|
| $\lambda_{\text{time}} = 3.5$ | [0.67; **0.80**; 1.34] | [0.66; **0.70**; 0.76] |
| $\lambda_{\text{time}} = 50$ | [0.71; **0.88**; 1.13] | [0.69; **0.72**; 0.74] |

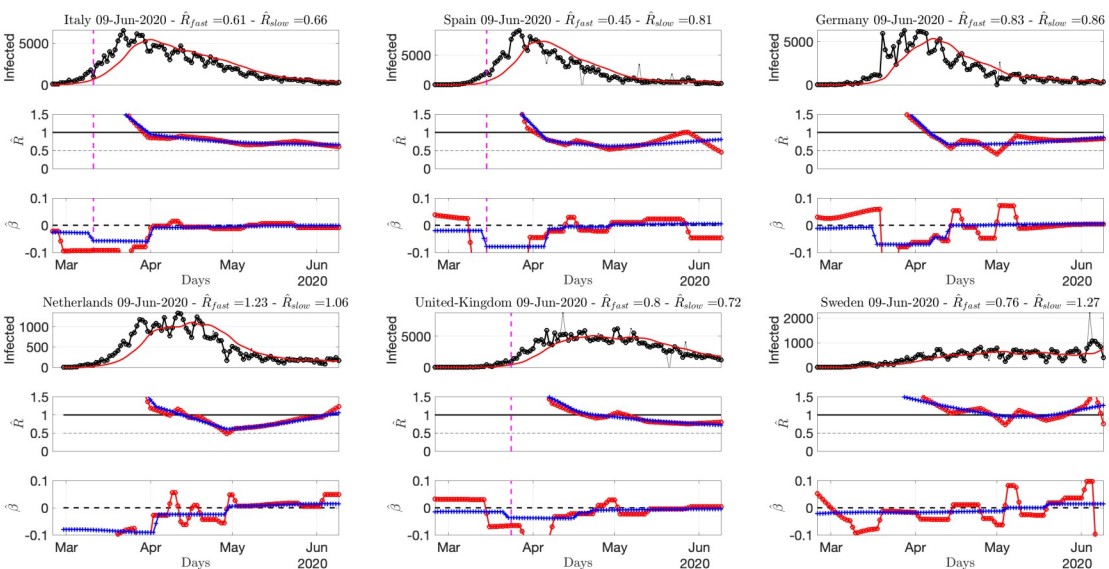

**Fig 5. Number of daily new confirmed cases for Europe, reproduction numbers and local trends.** Top: time series. Middle: fast (red) and slowly evolving (blue) estimates of $R(t)$. Bottom: fast (red) and slowly evolving (blue) estimates of local trends $\beta(t)$. The title of the plots reports the slow and fast estimates of $R$ for the current day. Data from Source2(ECDPC).

*Phase-space representation.* To complement Figs 5 to 8, Fig 9 displays a phase-space representation of the time evolution of the pandemic, constructed by plotting one against the other the local average (over a week) of the slowly varying estimated reproduction number $\hat{R}(t)$ and local trend, $(\bar{R}(t), \bar{\beta}(t))$, for a period ranging from mid-April to June 9th. Country names are written at the end (last day) of the trajectories.

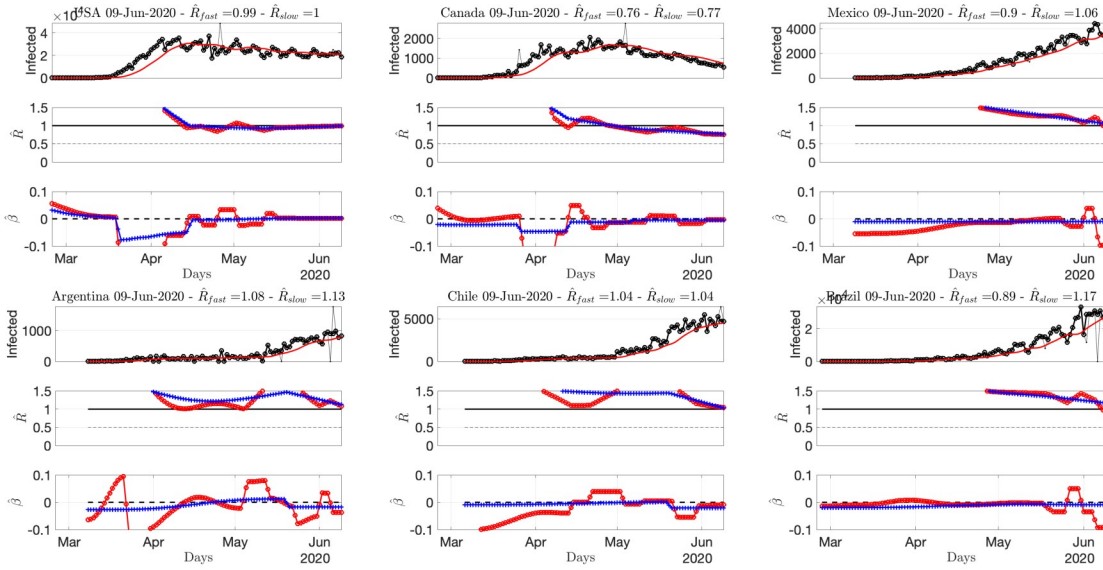

**Fig 6. Number of daily new confirmed cases for American countries, reproduction numbers and local trends.** Top: time series. Middle: fast (red) and slowly evolving (blue) estimates of $R(t)$. Bottom: fast (red) and slowly evolving (blue) estimates of local trends $\beta(t)$. The title of the plots reports the slow and fast estimates of $R$ for the current day. Data from Source2(ECDPC).

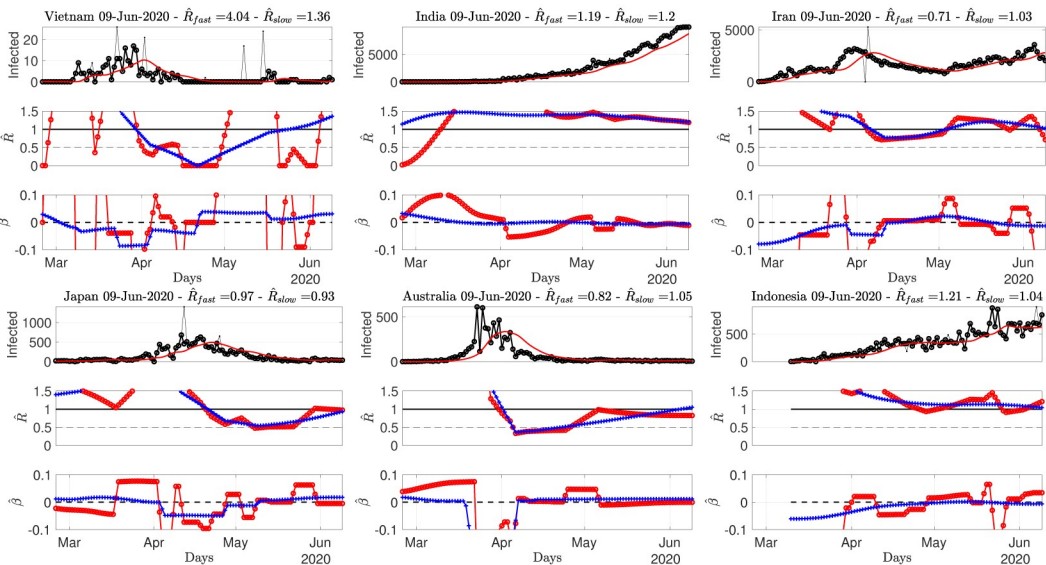

**Fig 7. Number of daily new confirmed cases for Asian countries, reproduction numbers and local trends.** Top: time series. Middle: fast (red) and slowly evolving (blue) estimates of $R(t)$. Bottom: fast (red) and slowly evolving (blue) estimates of local trends $\beta(t)$. The title of the plots reports the slow and fast estimates of $R$ for the current day. Data from Source2 (ECDPC).

Interestingly, European countries display a C-shape trajectory, starting with $R > 1$ with negative trends (lockdown effects), thus reaching the safe zone ($R < 1$) but eventually performing a U-turn with a slow increase of local trends till positive. This results in a mild but clear re-increase of $R$, yet with most values below 1 today, except for France (see comments above) and Sweden. The USA display a similar C-shape though almost concentrated on the edge

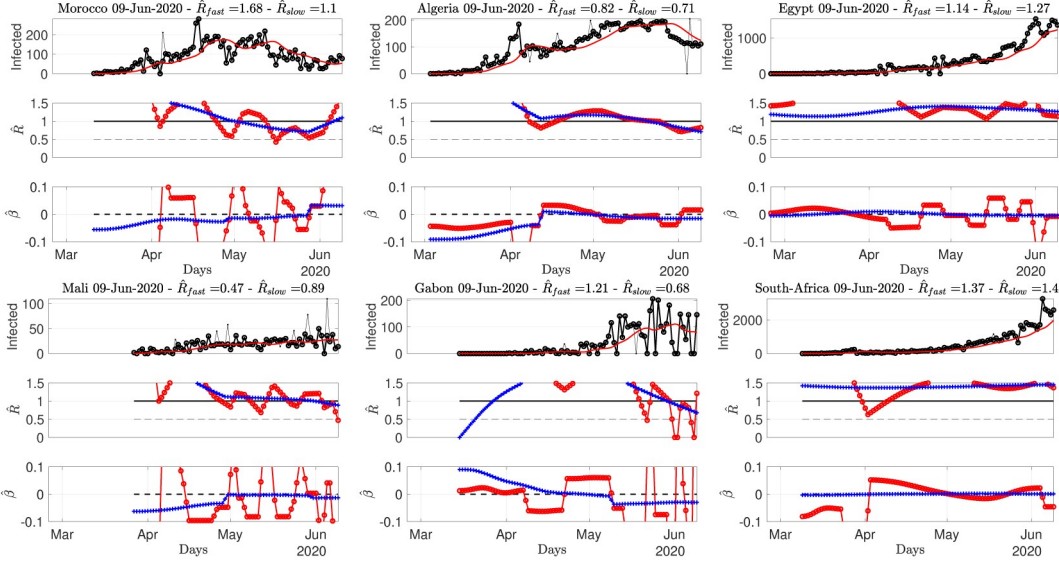

**Fig 8. Number of daily new confirmed cases for African countries, reproduction numbers and local trends.** Top: time series. Middle: fast (red) and slowly evolving (blue) estimates of $R(t)$. Bottom: fast (red) and slowly evolving (blue) estimates of local trends $\beta(t)$. The title of the plots report estimates for the current day. Data from Source2(ECDPC).

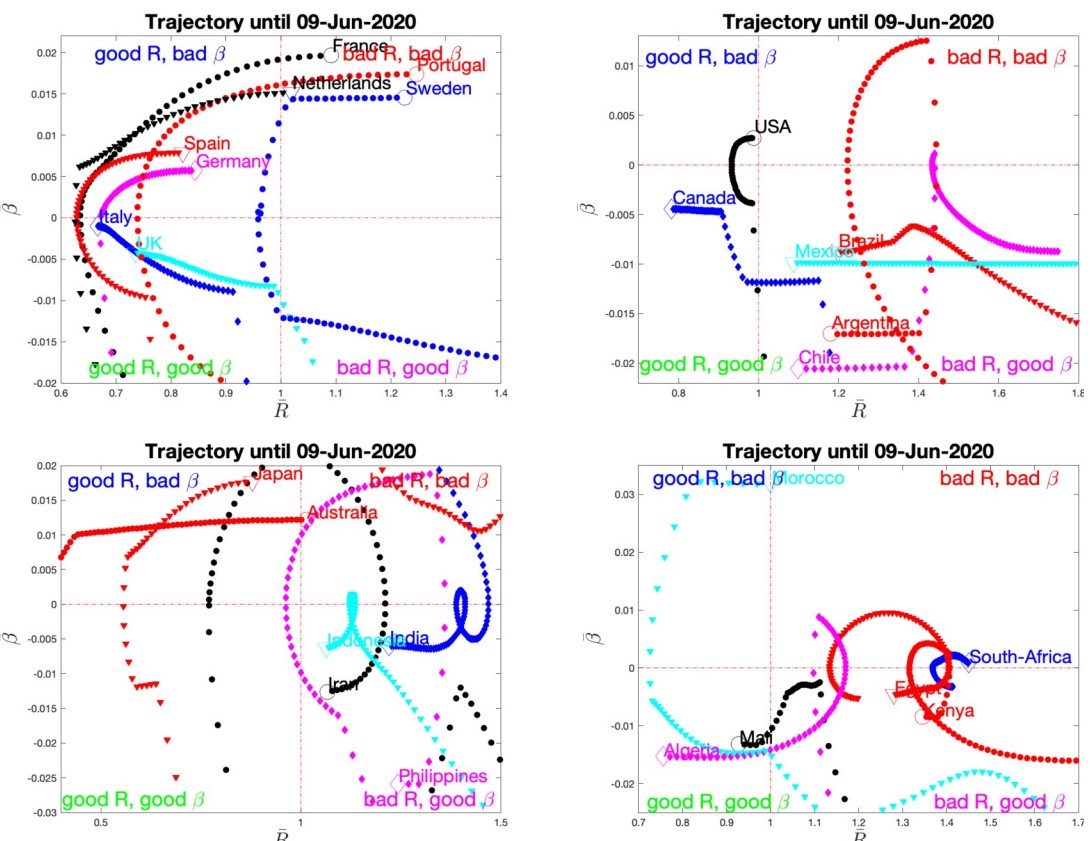

**Fig 9. Phase-space evolution reconstructed from averaged slowly varying estimates of *R* and *β*, per continent.** The name of the country is written at the last day of the trajectory, also marked by larger size empty symbol. Data from Source2(ECDPC).

point $R(t) = 1, \beta = 0$, while Canada does return to the safe zone with a specific pattern. South-American countries, obviously at an earlier stage of the pandemic, show an inverted C-shape pattern, with trajectory evolving from the bad top right corner, to the *controlling phase* (negative local trend, with decreasing *R* still above 1 though). Phase-spaces of Asian and African countries essentially confirm these C-shaped trajectories.

Envisioning these phase-space plots as pertaining to different stages of the pandemic (rather than to different countries), this suggests that COVID-19 pandemic trajectory resembles a clockwise circle, starting from the *bad* top right corner (*R* above 1 and positive trends), evolving, likely by lockdown impact, towards the bottom right corner (*R* still above 1 but negative trends) and finally to the *safe* bottom left corner (*R* below 1 and negative then null trend). The lifting of the lockdown may explain the continuation of the trajectory in the *still safe but. . .* corner (*R* below1 and again positive trend). As of June 9th, it can be only expected that trajectories will not close the loop and reach back the *bad* top right corner and the *R* = 1 limit.

**Continental France *départements*: Regularized joint estimates.** There is further interest in focusing the analysis on the potential heterogeneity in the epidemic propagation across a given territory, governed by the same sanitary rules and health care system. This can be achieved by estimating a set of *local* $\hat{R}(t)$'s for different provinces and regions [5].

Such a study is made possible by the data from Source3(SPF), that provides hospital-based data for each of the continental France *départements*. Fig 4 (right) already reported the slow and fast varying estimates of *R* and local trends computed from data aggregated over the whole

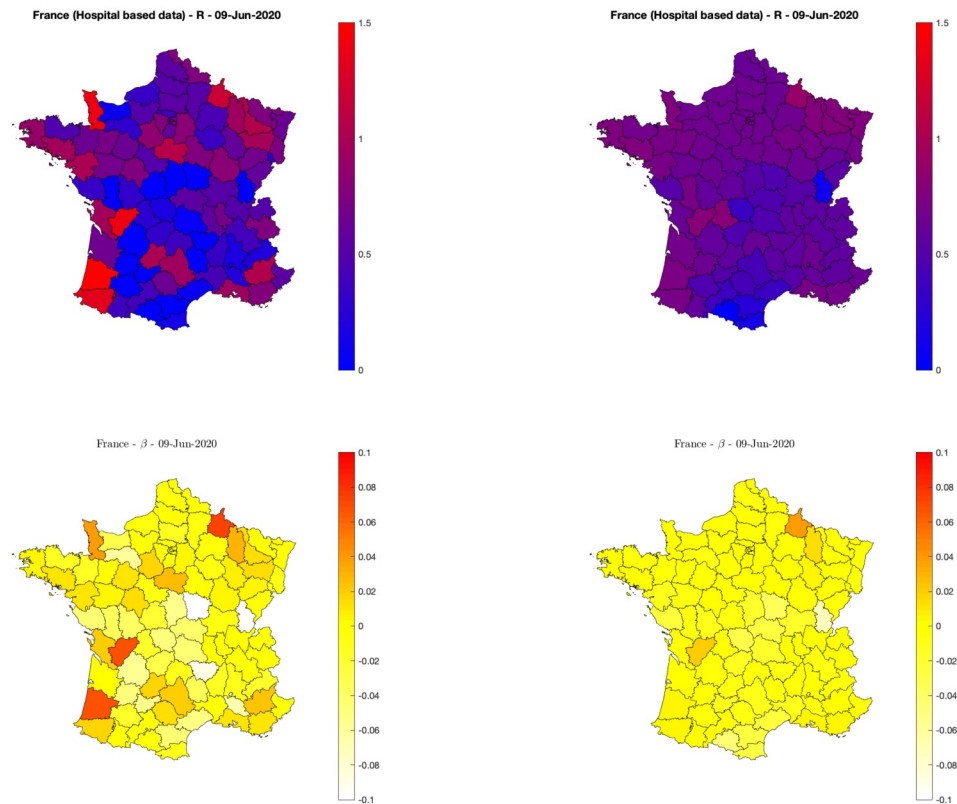

**Fig 10. Reproduction numbers and trends for continental France *départements*.** Fast varying estimates of reproduction numbers *R* (top) and trends *β* (bottom) for independent (left) and spatial graph-based regularized estimates (right). Hospital-based data from Source3(SPF). MapData©OpenStreetMap contributors.

France. To further study the variability across the continental France territory, the graph-based, joint spatial and temporal regularization described in Eq 7 is applied to the number of confirmed cases consisting of a matrix of size $K \times T$, with $D = 94$ continental France *départements*, and $T$ the number of available daily data (e.g., $T = 78$ on June 9th, data being available only after March 18th). The choice $\lambda_{\text{time}} = 3.5$ leading to fast estimates was used for this joint study. Using (10) as a guideline, empirical analyses led to set $\lambda_{\text{space}} = 0.025$, thus selecting spatial regularization to weight one-fourth of the temporal regularization.

First, Fig 10 (top row) maps and compares for June 9th (chosen arbitrarily as the day of writing) per-*département* estimates, obtained when *départements* are analyzed either independently ($\hat{R}_{\text{Indep}}$ using Eq 6, left plot) or jointly ($\hat{R}_{\text{Joint}}$ using Eq 7, right plot). While the means of $\hat{R}_{\text{Indep}}$ and $\hat{R}_{\text{Joint}}$ are of the same order ($\simeq 0.58$ and $\simeq 0.63$ respectively) the standard deviations drop down from $\simeq 0.40$ to $\simeq 0.14$, thus indicating a significant decrease in the variability across departments. This is further complemented by the visual inspection of the maps which reveals reduced discrepancies across neighboring departments, as induced by the estimation procedure.

In a second step, short and long-term trends are automatically extracted from $\hat{R}_{\text{Indep}}$ and $\hat{R}_{\text{Joint}}$ and short-term trends are displayed in the bottom row of Fig 10 (left and right, respectively). This evidences again a reduced variability across neighboring departments, though much less than that observed for $\hat{R}_{\text{Indep}}$ and $\hat{R}_{\text{Joint}}$, likely suggesting that trends on $R$ per se are

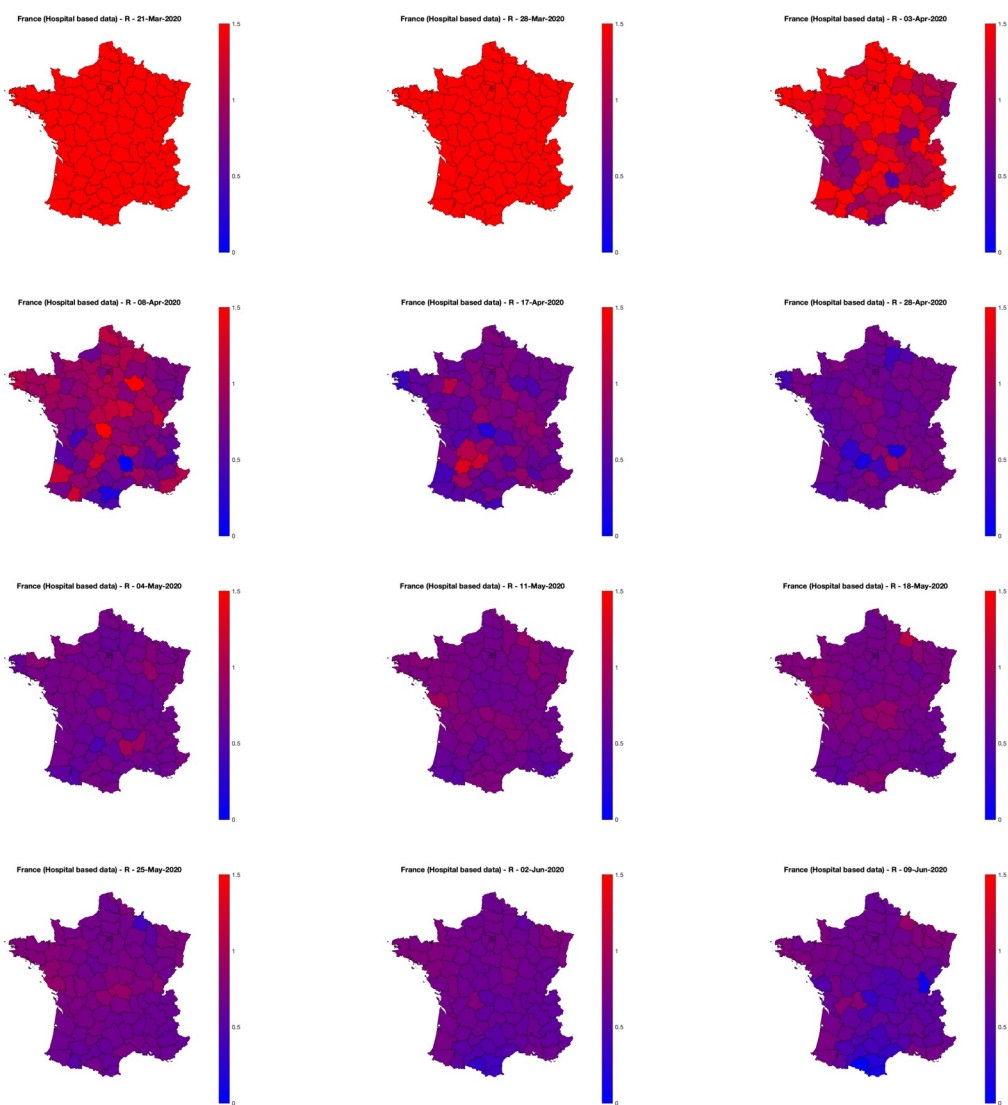

**Fig 11. Graph-based spatially regularized estimates of the reproduction number *R* for the 94 continental France *départments*, as a function of days.** Movie animations for the whole period are made available at perso.ens-lyon.fr/patrice.abry/DeptRegul.mp4 or barthes.enssib.fr/coronavirus/IXXI-SiSyPhe/, and updated on a regular basis. Hospital-based data from Source3 (SPF). MapData©OpenStreetMap contributors.

more robust quantities to estimate than single *R*'s. For June 9th, Fig 10 also indicates reproduction numbers that are essentially stable everywhere across France, thus confirming the trend estimated on data aggregated over all France (cf. Fig 4, right plot).

Video animations, available at perso.ens-lyon.fr/patrice.abry/DeptRegul.mp4, and at barthes.enssib.fr/coronavirus/IXXI-SiSyPhe/., updated on a daily basis, report further comparisons between $\hat{R}_{\mathrm{Indep}}$ and $\hat{R}_{\mathrm{Joint}}$ and their evolution along time for the whole period of data availability. Maps for selected days are displayed in Fig 11 (with identical colormaps and colorbars across time). Fig 11 shows that until late March (lockdown took place in France on March 17th), $\hat{R}_{\mathrm{Joint}}$ was uniformly above 1.5 (chosen as the upper limit of the colorbar to permit to see variations during the lockdown and post-lockdown periods), indicating a rapid evolution of the epidemic across entire France. A slowdown of the epidemic evolution is visible as early as

the first days of April (with overall decreases of $\hat{R}_{\text{Joint}}$, and a clear North vs. South gradient). During April, this gradient rotates slightly and aligns on a North-East vs. South-West direction and globally decreases in amplitude. Interestingly, in May, this gradient has reversed direction from South-West to North-East, though with very mild amplitude. As of today (June 9th), the pandemic, viewed Hospital-based data from Source3(SPF), seems under control under the whole continental France.

## Conclusion

### Discussion

Estimation of the reproduction number constitutes a classical task in assessing the status of a pandemic. Classically, this is done a posteriori (after the pandemic) and from consolidated data, often relying on detailed and accurate SIR-based models and relying on Bayesian frameworks for estimation. However, on-the-fly monitoring of the reproduction number time evolution constitutes a critical societal stake in situations such as that of COVID-19, when decisions need to be taken and action need to be made under emergency. This calls for a triplet of constraints: i) robust access to fast-collected data; ii) semi-parametric models for such data that focus on a subset of critical parameters; iii) estimation procedures that are both elaborated enough to yield robust estimates, and versatile enough to be used on a daily basis and applied to (often-limited in quality and quantity) available data.

In that spirit, making use of a robust nonstationary Poisson-distribution based semi-parametric model proven robust in the literature for epidemic analysis, we developed an original estimation procedure to favor piecewise regular estimation of the evolution of the reproduction number, both along time and across space. This was based on an inverse problem formulation balancing fidelity to time and space regularization, and used proximal operators and nonsmooth convex optimization.

This tool can be applied to time series of incidence data, reported, e.g., for a given country. Whenever made possible from data, estimation can benefit from a graph of spatial proximity between subdivisions of a given territory. The tool also provides local trends that permit to forecast short-term future values of *R*.

The proposed tools were applied to pandemic incidence data consisting of daily counts of new infections, from several databases providing data either worldwide on an aggregated per-country basis or, for France only, based on the sole hospital counts, spread across the French territory. They permitted to reveal interesting patterns on the state of the pandemic across the world as well as to assess variability across one single territory governed by the same (health care and politics) rules. More importantly, these tools can be used everyday easily as an on-the-fly monitoring procedure for assessing the current state of the pandemic and predict its short-term future evolution.

### Updates and software tools

Updated estimations are published on-line every day at perso.ens-lyon.fr/patrice.abry and at barthes.enssib.fr/coronavirus/IXXI-SiSyPhe/. Data were (and still are) automatically downloaded on a daily basis using routines written by ourselves. All tools have been developed in MATLAB™ and can be made available from the corresponding author upon motivated request.

### Future works

At the methodological level, the tool can be further improved in several ways. Instead of using $\Omega(\mathbf{R}) \coloneqq \lambda_{\text{time}}\|\mathbf{D}_2\,\mathbf{R}\|_1 + \lambda_{\text{space}}\|\mathbf{R}\mathbf{B}^\top\|_1$, for the joint time and space regularization, another

possible choice is to directly consider the matrix $\mathbf{D}_2\,\mathbf{RB}^\top$ of joint spatio-temporal derivatives, and to promote sparsity with an $\ell_1$-norm, or structured sparsity with a mixed norm $\ell_{1,2}$, e.g., $\|\mathbf{D}_2\,\mathbf{RB}^\top\|_{1,2} = \Sigma_t\|(\mathbf{D}_2\,\mathbf{RB}^\top)(t)\|_2$.

As previously discussed, data collected in the process of a pandemic are prone to several causes for outliers. Here, outlier preprocessing and reproduction number estimation were conducted in two independent steps, which can turn suboptimal. They can be combined into a single step at the cost of increasing the representation space permitting to split observation in true data and outliers, by adding to the functional to minimize an extra regularization term and devising the corresponding optimization procedure, which becomes nonconvex, and hence far more complicated to address.

Finally, when an epidemic model suggests a way to make use of several time series (such as, e.g., infected and deceased) for one same territory, the tool can straightforwardly be extended into a multivariate setting by a mild adaptation of optimization problems (6) and (7), replacing the Kullback-Leibler divergence $D_{\mathrm{KL}}(\mathbf{Z}|\mathbf{R}\odot\mathbf{\Phi Z})$ by $\sum_{i=1}^{I} D_{\mathrm{KL}}(\mathbf{Z}^i\mid\mathbf{R}\odot\mathbf{\Phi Z}^i)$. Finally, automating a data-driven tuning of the regularization hyperparameters constitutes another important research track.

## Acknowledgments

Figs 10 and 11 are produced using open ressources from the OpenStreetMap foundation, whose contributors are here gratefully acknowledged. *MapData©OpenStreetMap contributors.*

## Author Contributions

**Conceptualization:** Patrice Abry, Pablo Jensen, Patrick Flandrin.

**Data curation:** Stéphane Roux.

**Formal analysis:** Patrice Abry, Nelly Pustelnik, Charles-Gérard Lucas.

**Investigation:** Patrice Abry, Pablo Jensen, Patrick Flandrin, Pierre Borgnat, Nicolas Garnier.

**Methodology:** Patrice Abry, Nelly Pustelnik, Rémi Gribonval.

**Resources:** Patrice Abry.

**Software:** Patrice Abry, Nelly Pustelnik, Charles-Gérard Lucas.

**Supervision:** Patrice Abry.

**Validation:** Patrice Abry.

**Visualization:** Stéphane Roux, Éric Guichard.

**Writing – original draft:** Patrice Abry, Pablo Jensen, Patrick Flandrin, Rémi Gribonval.

**Writing – review & editing:** Patrice Abry, Pablo Jensen, Patrick Flandrin.

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
