## [Decision Letter · Decision Letter 0]

28 Jul 2020

PONE-D-20-21676

Spatial and temporal regularization to estimate COVID-19 Reproduction Number R(t): 

Promoting piecewise smoothness via convex optimization

PLOS ONE

Dear Dr. Abry,

Thank you for submitting your manuscript to PLOS ONE. After careful consideration, we feel that it has merit but does not fully meet PLOS ONE’s publication criteria as it currently stands. Therefore, we invite you to submit a revised version of the manuscript that addresses the points raised during the review process.

We look forward to receiving your revised manuscript.

Kind regards,

Maria Alessandra Ragusa, PhD Professor

Academic Editor

PLOS ONE

Journal Requirements:

2.We note that you have provided URLs to the datasets used within your Methods section. We ask that you additionally provide these URLs in the Data Availability statement.

PLOS ONE does not allow for the use of footnotes in its publications. As such, we ask that you move the information provided in the footnotes to the main text.

3.PLOS requires an ORCID iD for the corresponding author in Editorial Manager on papers submitted after December 6th, 2016. Please ensure that you have an ORCID iD and that it is validated in Editorial Manager. To do this, go to ‘Update my Information’ (in the upper left-hand corner of the main menu), and click on the Fetch/Validate link next to the ORCID field. This will take you to the ORCID site and allow you to create a new iD or authenticate a pre-existing iD in Editorial Manager. Please see the following video for instructions on linking an ORCID iD to your Editorial Manager account: https://www.youtube.com/watch?v=_xcclfuvtxQ

4. Please amend the manuscript submission data (via Edit Submission) to include author Charles-Gerard Lucas.

5.We note that [Figure(s) 10, 11] in your submission contain [map/satellite] images which may be copyrighted. All PLOS content is published under the Creative Commons Attribution License (CC BY 4.0), which means that the manuscript, images, and Supporting Information files will be freely available online, and any third party is permitted to access, download, copy, distribute, and use these materials in any way, even commercially, with proper attribution. For these reasons, we cannot publish previously copyrighted maps or satellite images created using proprietary data, such as Google software (Google Maps, Street View, and Earth). For more information, see our copyright guidelines: http://journals.plos.org/plosone/s/licenses-and-copyright.

1.    You may seek permission from the original copyright holder of Figure(s) [10, 11] to publish the content specifically under the CC BY 4.0 license. 

<h1>** **</h1>

Additional Editor Comments (if provided):

The paper, after minor revision according to the attached report, could be accepted for publication.

Reviewers' comments:

Reviewer's Responses to Questions

**Comments to the Author**

1. Is the manuscript technically sound, and do the data support the conclusions?

Reviewer #1: Yes

2. Has the statistical analysis been performed appropriately and rigorously? 

Reviewer #1: Yes

3. Have the authors made all data underlying the findings in their manuscript fully available?

Reviewer #1: Yes

4. Is the manuscript presented in an intelligible fashion and written in standard English?

Reviewer #1: Yes

5. Review Comments to the Author

Reviewer #1: PLOS ONE

Manuscript Number: PONE-D-20-21676

Manuscript Title: Spatial and temporal regularization to estimate COVID-19 Reproduction Number R(t): Promoting piecewise smoothness via convex optimization

This is a really interesting paper.

The study presents the results of original research. The paper is interesting and, useful both in a mathematical sense and in medical one.

The author suggested a new estimation procedure to quantify the spread of a new epidemic in order to permit the monitoring of the evolution of the reproduction number.

Research work is motivated by the need to find a new method to monitoring the pandemic evolution in real time. The work proposed a new framework for the estimation of R(t).

Incidence data referes to the number of daily new infections in several conutries and secondarly in France departments.

1. INTRODUCTION

Context

To enrich the work of interest it might be useful to explain the practical usefulness of having new predictive models inserting some sentences such as:

“The ability to provide a predictive model is extremely useful for management of patients and also for medical resource allocation. What was in fact achieved in recent months of February and March was the difficulty to have models that could predict the number of patients (not only the infected) and what was the real need for personal protection and support for patients in hospitals.

At that time, in fact, the data and methods of spreading the epidemic in China were not fully available, but already in Europe, primarily in Italy, the epidemic was spreading. In fact, we were not talking about a pandemic yet and we were not able to meet the needs of gloves, disinfectant, facial mask, mechanical ventilators and beds of intensive care”. The goal of the entire research community is to find a solution to provide timely data to prevent future health crises.

To support these sentence authors can cite these COVID-19 update articles or others that they can choose:

Coccia M.

Factors determining the diffusion of COVID-19 and suggested strategy to prevent future accelerated viral infectivity similar to COVID.

Sci Total Environ. 2020 Aug 10;729:138474. doi: 10.1016/j.scitotenv.2020.138474.

Petkova E, Antman EM, Troxel AB.

Pooling Data From Individual Clinical Trials in the COVID-19 Era

JAMA. Published online July 22, 2020. doi:10.1001/jama.2020.13042

Ing, RJ; Bills, C; Merritt, G et al.

Role of Helmet-Delivered Noninvasive Pressure Support Ventilation in COVID-19 Patients.

J Cardiothorac Vasc Anesth; 2020 May 08. doi: 10.1053/j.jvca.2020.04.060.

The part “issue and related work” is too long: it is recommended to eliminate the details that will be described later to better focus the work points.

MATHERIALS AND METHODS

To make the reading clearer it is necessary to delete parts already described or comments and just describe the data.

Model and estimation procedures represent the core of the work and unnecessary or repeated parts should be eliminated (for example line 125-127 “to enable..basis”or line 151-152 “outliers…ways” or line 167-169 “as…outbreak”).

RESULTS

The study presents the results of original research.

Results reported have not been published elsewhere.

Methods are described in sufficient detail for another researcher to reproduce the experiments described.

CONCLUSIONS

In order to clarify the conclusions it is necessary to delete some phrases or concepts already described (for example line 520-532 “Estimation….analysis”; line 546-552 “As previously… address”).

I suggest to move the sentence “when decisions need to be taken and action need to be made 525

under emergency” (line 525-526) at the end of the conclusion paragraph (line 566).

6. PLOS authors have the option to publish the peer review history of their article (what does this mean?). If published, this will include your full peer review and any attached files.

Reviewer #1: No

---

## [Author Response · Author response to Decision Letter 0]

31 Jul 2020

Responses to The Editor and Reviewer were given in the cover letter file (Coverletter.pdf) and in the responses to reviewer file (responses2reviewer.pdf)

---

## [Editor Report · Decision Letter 1]

6 Aug 2020

Spatial and temporal regularization to estimate COVID-19 Reproduction Number R(t): 

Promoting piecewise smoothness via convex optimization

PONE-D-20-21676R1

Dear Dr. Abry,

We’re pleased to inform you that your manuscript has been judged scientifically suitable for publication and will be formally accepted for publication once it meets all outstanding technical requirements.

Kind regards,

Maria Alessandra Ragusa, PhD Professor

Academic Editor

PLOS ONE
---

## [Editor Report · Acceptance letter]

7 Aug 2020

PONE-D-20-21676R1 

Spatial and temporal regularization to estimate COVID-19 Reproduction Number R(t): 
Promoting piecewise smoothness via convex optimization 

Dear Dr. Abry:

I'm pleased to inform you that your manuscript has been deemed suitable for publication in PLOS ONE. Congratulations! Your manuscript is now with our production department. 

Kind regards, 

on behalf of

Dr. Maria Alessandra Ragusa 

Academic Editor

PLOS ONE